# GRAC: Self-Guided and Self-Regularized Actor-Critic

**Lin Shao[1], Yifan You[2], Mengyuan Yan[1], Shenli Yuan[1], Qingyun Sun[3], Jeannette Bohg[1]**

[1]Stanford AI Lab, [3]Department of Mathematics, Stanford University
{lins2, mengyuan, shenliy, qysun, bohg}@stanford.edu
[2]Department of Computer Science, University of California, Los Angeles
harry473417@ucla.edu
*

**Abstract:** Deep reinforcement learning (DRL) algorithms have successfully been demonstrated on a range of challenging decision making and control tasks. One dominant component of recent deep reinforcement learning algorithms is the target network which mitigates the divergence when learning the Q function. However, target networks can slow down the learning process due to delayed function updates. Our main contribution in this work is a self-regularized TD-learning method to address divergence without requiring a target network. Additionally, we propose a self-guided policy improvement method by combining policy-gradient with zero-order optimization to search for actions associated with higher Q-values in a broad neighborhood. This makes learning more robust to local noise in the Q function approximation and guides the updates of our actor network. Taken together, these components define *GRAC*, a novel self-guided and self-regularized actor critic algorithm. We evaluate *GRAC* on the OpenAI gym tasks, outperforming popular methods such as *TD3* [1] and *SAC* [2] on four tasks and achieving competitive results on two environments. We also apply *GRAC* to enable a non-anthropomorphic robotic hand to successfully accomplish an in-hand manipulation task in the real world.

**Keywords:** Deep Reinforcement Learning, Q-learning

## 1 Introduction

Reinforcement learning (RL) studies decision-making with the goal of maximizing total discounted reward when interacting with an environment. Leveraging high-capacity function approximators such as neural networks, Deep reinforcement learning (DRL) algorithms have been successfully applied to a range of challenging domains, from video games [3] to robotic control [4].

Actor-critic algorithms are among the most popular approaches in DRL, e.g. *DDPG* [5], *TRPO* [4], *TD3* [1] and *SAC* [2]. These methods are based on policy iteration, which alternates between policy evaluation and policy improvement [6]. Actor-critic methods jointly optimize the value function (critic) and the policy (actor) as it is often impractical to run either of these to convergence [2].

In DRL, both the actor and critic use deep neural networks as the function approximator. However, DRL is known to assign unrealistically high values to state-action pairs represented by the Q-function. This is detrimental to the quality of the greedy control policy derived from Q [7]. Mnih et al. [8] proposed to use a *target network* to mitigate divergence. A target network is a copy of the current Q function that is held fixed to serve as a stable target within the TD error update. The parameters of the target network are either infrequently copied [8] or obtained by Polyak averaging [5]. A limitation of using a target network is that it can slow down learning due to delayed function updates. We propose

---

*Toyota Research Institute ("TRI") provided funds to assist the authors with their research but this article solely reflects the opinions and conclusions of its authors and not TRI or any other Toyota entity. This article is also supported by Siemens and the HAI-AWS Cloud Credits.

5th Conference on Robot Learning (CoRL 2021), London, UK.

an approach that reduces the need for a target network in DRL while still ensuring stable learning and good performance in high-dimensional domains. We add a self-regularization term to encourage small changes to the target value while minimizing the *Temporal Difference* (TD)-error [6].

*Evolution Strategies* (ES) are a family of black-box optimization algorithms which are typically very stable, but scale poorly in high-dimensional search spaces e.g. neural networks [9]. Policy gradient-based DRL methods, unlike evolutionary search methods, can continue to sample previous experiences to improve value estimation, particularly in the off-policy setting. At the same time, these approaches can also be unstable and highly sensitive to hyper-parameter tuning [9]. We propose a novel policy improvement method which combines both approaches to get the best of both worlds. Specifically, after the actor network first outputs an initial action, we apply the *Cross Entropy Method* (CEM) [10] to search the neighborhood of the initial action to find a second action associated with a higher Q value. Then we leverage the second action in the policy improvement stage to speed up the learning process.

Our main contribution in this work is a **self-regularized** TD-learning method to address divergence without requiring a target network that may slow down learning progress. In addition, we propose a **self-guided** policy improvement method which combines policy-gradients and zero-order optimization. This helps to speed up learning and is robust to local noise in the Q function approximation. Taken together, these components define *GRAC*, a novel self-guided and self-regularized actor critic algorithm. We evaluate *GRAC* on six continuous control domains from OpenAI gym [11], where *GRAC* outperforms popular methods such as *TD3* [1] and *SAC* [2] on four tasks and achieves competitive results on two environments. We run our experiments across a large number of seeds with fair evaluation metrics [12], perform extensive ablation studies, and open source both our code and learning curves. We also run *GRAC* to enable a non-anthropomorphic, real robotic hand to successfully rotate a cube to the target pose.

## 2 Related Work

The proposed algorithm incorporates key ingredients within the actor-critic method: a self-regularized TD update and self-guided policy improvements based on evolution strategies. In this section, we review prior work related to these ideas.

**Divergence in Deep Q-Learning** In Deep Q-Learning, we use a nonlinear function approximator such as a neural network to approximate the Q-function that represents the value of each state-action pair. Learning the Q-function in this way is known to suffer from divergence issues [13] such as assigning unrealistically high values to state-action pairs [7]. For the case when the control policy is greedily derived from Q [7], unrealistically high Q values are detrimental. To mitigate the divergence issue, Mnih et al. [8] introduce a target network which is a copy of the estimated Q-function and is held fixed to serve as a stable target for some number of steps. However, because of the delayed function updates, target networks can slow down learning [14]. Durugkar and Stone [15] propose Constrained Q-Learning, which uses a constraint to prevent the average target value from changing after an update. Achiam et al. [16] give a simple analysis based on a linear approximation of the Q function and develop a stable Deep Q-Learning algorithm for continuous control without target networks. However, their proposed method requires separately calculating backward passes for each state-action pair in the batch, and solving a system of equations at each timestep. Bhatt et al. [17] introduce a normalization called cross-normalization which is regarded as an extension of batch normalization that re-centers data for on- and off-policy transitions. Peng et al. [18] uses Monte Carlo instead of TD error to update the Q network and removes the need for a target network. Our proposed *GRAC* algorithm adds a self-regularization term to the TD-Learning objective to keep the change of the state-action value small.

**Evolution Strategies in Deep Reinforcement Learning** Evolution Strategies are typically stable but suffer from scaling to high-dimensional search spaces. Policy gradient-based deep RL methods, such as *DDPG* [5] can continue to reuse previous experience to improve value estimations, particularly in the off-policy setting, but can be unstable and highly sensitive to hyper-parameter tuning [9]. Researchers have proposed to combine these approaches to get the best of both worlds. Pourchot and Sigaud [9] proposed *CEM-RL* to combine CEM with either *DDPG* [5] or *TD3* [1]. However, *CEM-RL* applies *CEM* within the actor parameter space which is extremely high-dimensional, making

the search not efficient. Kalashnikov et al. [19] introduce *QT-Opt*, which leverages *CEM* to search the landscape of the Q function, and enables Q-Learning in continuous action spaces without using an actor. Based on *QT-Opt*, Simmons-Edler et al. [20] leverage CEM to search the landscape the Q-function but propose to initialize CEM with a Gaussian prior covering the action space, independent of observations. However, *CEM* does not scale well to high-dimensional action spaces [21], such as in the Humanoid task. We first let the actor network output an initial Gaussian action distribution conditioned on current state. Then we use CEM to search for an action with a higher Q value than the Q value of the Gaussian mean. Starting from the predicted distribution, we show that *GRAC* speeds up the learning process compared to popular actor-critic methods.

## 3 Preliminaries

In this section, we define the notation used in subsequent sections. Consider a *Markov Decision Process* (MDP), defined by the tuple $(\mathcal{S}, \mathcal{A}, \mathcal{P}, r, \rho_0, \gamma)$, where $\mathcal{S}$ is a finite set of states, $\mathcal{A}$ is a finite set of actions, $\mathcal{P} : \mathcal{S} \times \mathcal{A} \times \mathcal{S} \to \mathbb{R}$ is the transition probability distribution, $r : \mathcal{S} \times \mathcal{A} \to \mathbb{R}$ is the reward function, $\rho_0 : \mathcal{S} \to \mathbb{R}$ is the distribution of the initial state $s_0$, and $\gamma \in [0, 1]$ is the discount factor. At each discrete time step $t$, with a given state $s_t \in \mathcal{S}$, the agent selects an action $a_t \in \mathcal{A}$, receiving a reward $r$ and the new state $s_{t+1}$ of the environment.

Let $\pi$ denote the policy which maps a state to a probability distribution over the actions, $\pi : \mathcal{S} \to \mathcal{P}(\mathcal{A})$. The return from a state is defined as the sum of discounted reward $R_t = \sum_{i=t} \gamma^{i-t} r(s_i, a_i)$. In reinforcement learning, the objective is to find the optimal policy $\pi^*$, with parameters $\phi$, which maximizes the expected return $J(\phi) = \sum_t \mathbb{E}_{(s_t, a_t) \sim \rho_\pi(s_t, a_t)}[\gamma^t r(s_t, a_t)]$ where $\rho_\pi(s_t)$ and $\rho_\pi(s_t, a_t)$ denote the state and state-action marginals of the trajectory distribution induced by the policy $\pi(a_t|s_t)$.

We use the following standard definitions of the state-action value function $Q_\pi$. It describes the expected discounted reward after taking an action $a_t$ in state $s_t$ and thereafter following policy $\pi$:

$$Q_\pi(s_t, a_t) = \mathbb{E}_\pi[R_t | s_t, a_t].$$

In this work we use *CEM* to find optimal actions with maximum Q values. *CEM* is a randomized zero-order optimization algorithm. To find the action $a$ that maximizes $Q(s, a)$, *CEM* is initialized with a paramaterized distribution over $a$, $P(a; \psi)$. Then it iterates between the following two steps [22]: First generate $a_1, \ldots, a_N \sim P(s; \psi)$. Retrieve their Q values $Q(s, a_i)$ and sort the actions to have decreasing Q values. Then keep the first $K$ actions, and solve for updated parameters $\psi'$:

$$\psi' = \operatorname{argmax}_\psi \frac{1}{K} \sum_{i=1}^{K} \log(P(a_i; \psi))$$

In the following, let $CEM(Q(s, \cdot), \pi(\cdot|s))$ denote the action found by *CEM* to maximize $Q(s, \cdot)$, when *CEM* is initialized with the distribution predicted by the policy.

## 4 Technical Approach

### 4.1 Self-Regularized TD Learning

Reinforcement learning is prone to instability and divergence when a nonlinear function approximator such as a neural network is used to represent the Q function [13]. Mnih et al. [8] identified several reasons for this. One is the correlation between the current action-values and the target value. Updates to $Q(s_t, a_t)$ often also increase $Q(s_{t+1}, a_{t+1}^*)$ where $a_{t+1}^*$ is the optimal next action. Hence, these updates also increase the target value $y_t$ which may lead to oscillations or the divergence of the policy.

More formally, given transitions $(s_t, a_t, r_t, s_{t+1})$ sampled from the replay buffer distribution $\mathcal{B}$, the Q network can be trained by minimising the loss functions $\mathcal{L}(\theta_i)$ at iteration $i$:

$$\mathcal{L}(\theta_i) = \mathbb{E}_{(s_t, a_t) \sim \mathcal{B}} \|(Q(s_t, a_t; \theta_i) - y_i)\|^2 \tag{1}$$

where for now let us assume $y_i = r_t + \gamma \max_a Q(s_{t+1}, a; \theta_i)$ to be the target for iteration $i$ computed based on the current Q network parameters $\theta_i$. $a_{t+1}^* = \arg\max_a Q(s_{t+1}, a)$. If we update the parameter $\theta_{i+1}$ to reduce the loss $\mathcal{L}(\theta_i)$, it changes both $Q(s_t, a_t; \theta_{i+1})$ and $Q(s_{t+1}, a_{t+1}^*; \theta_{i+1})$. Assuming an increase in both values, then the new target value $y_{i+1} = r_t + \gamma Q(s_{t+1}, a_{t+1}^*; \theta_{i+1})$ for

---

**Algorithm 1** GRAC

---

Initialize critic network $Q_{\theta 1}$, $Q_{\theta 2}$ and actor network $\pi_\phi$ with random parameters $\theta 1$, $\theta 2$ and $\phi$
Initialize replay buffer $\mathcal{B}$, Set $\alpha < 1$

1: **for** $i = 1, \ldots$ **do**
2:      Select action $a \sim \pi_{\phi_i}(s)$, observe reward $r$ and new state $s'$
3:      Store transition tuple $(s, a, r, s')$ in $\mathcal{B}$
4:      Sample mini-batch of $N$ transitions $(s_t, a_t, r_t, s_{t+1})$ from $\mathcal{B}$.
5:      $\hat{a}_{t+1} \sim \pi_{\phi_i}(s_{t+1})$
6:      $\tilde{a}_{t+1} \leftarrow CEM(Q(s_{t+1}, \cdot; \theta_2), \pi_{\phi_i}(\cdot|s_{t+1}))$
7:      $y \leftarrow r_t + \gamma \max\{\min_{j=1,2} Q(s_{t+1}, \tilde{a}_{t+1}; \theta_j), \min_{j=1,2} Q(s_{t+1}, \hat{a}_{t+1}; \theta_j)\}$
8:      $a^\dagger \leftarrow \arg\max_{\{\tilde{a}, \hat{a}\}}\{\min_{j=1,2} Q(s_{t+1}, \tilde{a}_{t+1}; \theta_j), \min_{j=1,2} Q(s_{t+1}, \hat{a}_{t+1}; \theta_j)\}$
9:      $y'_1, y'_2 \leftarrow Q(s_{t+1}, a^\dagger; \theta_1), Q(s_{t+1}, a^\dagger; \theta_2)$
10:      **for** $k = 1$ to **K do**
11:          $\mathcal{L}_k = \|y - Q(s_t, a_t; \theta_1)\|_2^2 + \|y - Q(s_t, a_t; \theta_2)\|_2^2 + \|y'_1 - Q(s_{t+1}, a^\dagger; \theta_1)\|_2^2 + \|y'_2 - Q(s_{t+1}, a^\dagger; \theta_2)\|_2^2$
12:          $\theta 1 \leftarrow \theta 1 - \lambda \nabla_{\theta 1} \mathcal{L}_k$, $\theta 2 \leftarrow \theta 2 - \lambda \nabla_{\theta 2} \mathcal{L}_k$
13:          **if** $\mathcal{L}_k < \alpha \mathcal{L}_1$ **then**
14:             Break
15:          **end if**
16:      **end for**
17:      $\hat{a}_t \sim \pi_{\phi_i}(s_t)$
18:      $J_\pi(\phi) = \mathbb{E}_{(s_t, \hat{a}_t)}[Q(s_t, \hat{a}_t; \theta_1)]$
19:      $\bar{a}_t \leftarrow \text{CEM}(Q(s_t, \cdot; \theta_1), \pi_{\phi_i}(\cdot|s_t))$
20:      $\phi \leftarrow \phi - \lambda \nabla_\phi J_\pi(\phi) - \lambda \mathbb{E}_{(s_t, \hat{a}_t)}[Q(s_t, \bar{a}_t; \theta_1) - Q(s_t, \hat{a}_t; \theta_1)]_+ \nabla_\phi \log \pi(\bar{a}_t|s_t; \phi)$
21: **end for**

---

the next iteration will also increase leading to an explosion of the Q function. We demonstrated this behavior in an ablation experiment with results in Fig. 2. We also show how maintaining a separate target network [8] with frozen parameters $\theta^-$ to compute $y_{i+1} = r_t + \gamma Q(s_{t+1}, a^*_{t+1}; \theta^-)$ delays the update of the target and therefore leads to more stable learning of the Q function. However, delaying the function updates also comes with the price of slowing down the learning process.

We propose a self-Regularized TD-learning approach to minimize the TD-error while also keeping the change of $Q(s_{t+1}, a^*_{t+1})$ small. This regularization mitigates the divergence issue [13], and no longer requires a target network that would otherwise slow down the learning process. Let $y'_i = Q(s_{t+1}, a^*_{t+1}; \theta_i)$, and $y_i = r_t + \gamma y'_i$. We define the learning objective as

$$\min_\theta \|Q(s_t, a_t; \theta) - y_i\|^2 + \|Q(s_{t+1}, a^*_{t+1}; \theta) - y'_i\|^2 \tag{2}$$

where the first term is the original TD-Learning objective and the second term is the regularization term penalizing large updates to $Q(s_{t+1}, a^*_{t+1})$. Note that when the current Q network updates its parameters $\theta$, both $Q(s_t, a_t)$ and $Q(s_{t+1}, a^*_{t+1})$ change. Hence, the target value $y_i$ will also change which is different from the approach of keeping a frozen target network for a few iterations. We will demonstrate in our experiments that this self-regularized TD-Learning approach removes the delays in the update of the target value thereby achieves faster and stable learning.

### 4.2 Self-Guided Policy Improvement with Evolution Strategies

The policy, known as the actor, can be updated through a combination of two parts. The first part, which we call Q-loss policy update, improves the policy through local gradients of the current Q function, while the second part, which we call *CEM* policy update, finds a high-value action via *CEM* in a broader neighborhood of the Q function landscape and updates the action distribution to be around this high-value action. We formally describe the two parts below.

Given states $s_t$ sampled from the replay buffer, the Q-loss policy update maximizes the objective

$$J_\pi(\phi) = \mathbb{E}_{s_t \sim \mathcal{B}, \hat{a}_t \sim \pi}[Q(s_t, \hat{a}_t)], \tag{3}$$

where $\hat{a}_t$ is sampled from the current policy $\pi(\cdot|s_t)$. The gradient is taken through the reparameterization trick. We reparameterize the policy using a neural network transformation as described by Haarnoja et al. [2],

$$\hat{a}_t = f_\phi(\epsilon_t|s_t) \tag{4}$$

where $\epsilon_t$ is an input noise vector, sampled from a fixed distribution, such as a standard multivariate Normal distribution. Then the gradient of $J_\pi(\phi)$ is:

$$\nabla J_\pi(\phi) = \mathbb{E}_{s_t\sim\mathcal{B},\epsilon_t\sim\mathcal{N}}\left[\frac{\partial Q(s_t, f_\phi(\epsilon_t|s_t))}{\partial f}\frac{\partial f_\phi(\epsilon_t|s_t)}{\partial \phi}\right] \tag{5}$$

For the CEM policy update, given a minibatch of states $s_t$, we first find a high-value action $\bar{a}_t$ for each state by running *CEM* on the current Q function, $\bar{a}_t = CEM(Q(s_t, \cdot), \pi(\cdot|s_t))$. Then the policy is updated to increase the probability of this high-value action. The guided update on the parameter $\phi$ of $\pi$ at iteration $i$ is

$$\mathbb{E}_{s_t\sim\mathcal{B},\hat{a}_t\sim\pi}[Q(s_t,\bar{a}_t) - Q(s_t,\hat{a}_t)]_+\nabla_\phi \log \pi_i(\bar{a}_t|s_t). \tag{6}$$

We used $Q(s_t, \hat{a}_t)$ as a baseline term, since its expectation over actions $\hat{a}_t$ will give us the normal baseline $V(s_t)$:

$$\mathbb{E}_{s_t\sim\mathcal{B}}[Q(s_t,\bar{a}_t) - V(s_t)]_+\nabla_\phi \log \pi_i(\bar{a}_t|s_t) \tag{7}$$

In our implementation, we only perform an update if the improvement on the Q function, $Q(s_t,\bar{a}_t) - Q(s_t,\hat{a}_t)$, is non-negative, to guard against the occasional cases where *CEM* fails to find a better action.

Combining both parts, the final update rule on the parameter $\phi_i$ of policy $\pi_i$ is

$$\phi_{i+1} = \phi_i - \lambda\nabla_\phi J_{\pi_i}(\phi_i) - \lambda\,\mathbb{E}_{s_t\sim\mathcal{B},\hat{a}_t\sim\pi_i}[Q(s_t,\bar{a}_t)$$
$$-Q(s_t,\hat{a}_t)]_+\nabla_\phi \log \pi_i(\bar{a}_t|s_t)$$

where $\lambda$ is the step size.

Let $Q^\pi$ be the state-action value function induced by the current policy. We can prove that if the Q function has converged to $Q^\pi$ then both the Q-loss policy update and the *CEM* policy update will be guaranteed to improve the current policy. We formalize this result in Theorem 1 and Theorem 2, and prove them in Appendix.

**Theorem 1.** *Q-loss Policy Improvement Starting from the current policy $\pi$, we maximize the objective $J_\pi = \mathbb{E}_{(s,a)\sim\rho_\pi(s,a)} Q^\pi(s,a)$. The maximization converges to a critical point denoted as $\pi_{new}$. Then the induced Q function, $Q^{\pi_{new}}$, satisfies $\forall(s,a), Q^{\pi_{new}}(s,a) \geq Q^\pi(s,a)$.*

**Theorem 2.** CEM *Policy Improvement Assuming the* CEM *process is able to find the optimal action of the state-action value function, $a^*(s) = \arg\max_a Q^\pi(s,a)$, where $Q^\pi$ is the Q function induced by the current policy $\pi$. By iteratively applying the update $\mathbb{E}_{(s,a)\sim\rho_\pi(s,a)}[Q(s,a^*) - Q(s,a)]_+\nabla\log\pi(a^*|s)$, the policy converges to $\pi_{new}$. Then $Q^{\pi_{new}}$ satisfies $\forall(s,a), Q^{\pi_{new}}(s,a) \geq Q^\pi(s,a)$.*

### 4.3 Max-min Double Q-Learning

Q-learning [23] is known to suffer from overestimation [24]. Hasselt [25] proposed Double-Q learning which uses two Q functions with independent sets of weights to mitigate the overestimation problem. Fujimoto et al. [1] proposed Clipped Double Q-learning with two Q functions denoted as $Q(s,a;\theta_1)$ and $Q(s,a;\theta_2)$, or $Q_1$ and $Q_2$ in short. Given a transition $(s_t,a_t,r_t,s_{t+1})$, Clipped Double Q-learning uses the minimum between the two estimates of the Q functions when calculating the target value in TD-error [6]:

$$y = r_t + \gamma \min_{j=1,2} Q(s_{t+1}, \hat{a}_{t+1}; \theta_j) \tag{8}$$

where $\hat{a}_{t+1}$ is the predicted next action.

Fujimoto et al. [1] mentioned that such an update rule may induce an underestimation bias. In addition, $\hat{a}_{t+1} = \pi_\phi(s_{t+1})$ is the prediction of the actor network. The actor network's parameter $\phi$ is optimized according to the gradients of $Q_1$. In other words, $\hat{a}_{t+1}$ tends be selected according

to the $Q_1$ network which consistently increases the discrepancy between the two Q-functions. In practice, we observe that the discrepancy between the two estimates of the Q-function, $|Q_1 - Q_2|$, can increase dramatically leading to an unstable learning process.

We found that a technique we call *Max-min Double Q-Learning* reduces the discrepancy between the Q-functions. We first select $\hat{a}_{t+1}$ according to the actor network $\pi_\phi(s_{t+1})$. Then we run *CEM* to search the landscape of $Q_2$ within a broad neighborhood of $\hat{a}_{t+1}$ to return a second action $\tilde{a}_{t+1}$. Note that *CEM* selects an action $\tilde{a}_{t+1}$ that maximises $Q_2$ while the actor network selects an action $\hat{a}_{t+1}$ that maximises $Q_1$. We gather four different Q-values: $Q(s_{t+1}, \hat{a}_{t+1}; \theta_1)$, $Q(s_{t+1}, \hat{a}_{t+1}; \theta_2)$, $Q(s_{t+1}, \tilde{a}_{t+1}; \theta_1)$, and $Q(s_{t+1}, \tilde{a}_{t+1}; \theta_2)$. We then run a max-min operation to compute the target value that cancels the biases induced by $\hat{a}_{t+1}$ and $\tilde{a}_{t+1}$.

$$y = r_t + \gamma \max\{ \min_{j=1,2} Q(s_{t+1}, \hat{a}_{t+1}; \theta_j), \\ \min_{j=1,2} Q(s_{t+1}, \tilde{a}_{t+1}; \theta_j) \} \tag{9}$$

The inner min-operation $\min_{j=1,2} Q(s_{t+1}, \hat{a}_{t+1}; \theta_j)$ is adopted from Eq. 8 and mitigates overestimation [24]. The outer max operation helps to reduce the difference between $Q_1$ and $Q_2$. In addition, the max operation provides a better approximation of the Bellman optimality operator [6]. We visualize $Q_1$ and $Q_2$ during the learning process in the supplementary material. We formalize the convergence of the proposed Max-min Double Q-Learning approach in the finite MDP setting and prove this theorem in the Appendix.

# 5 Experiments

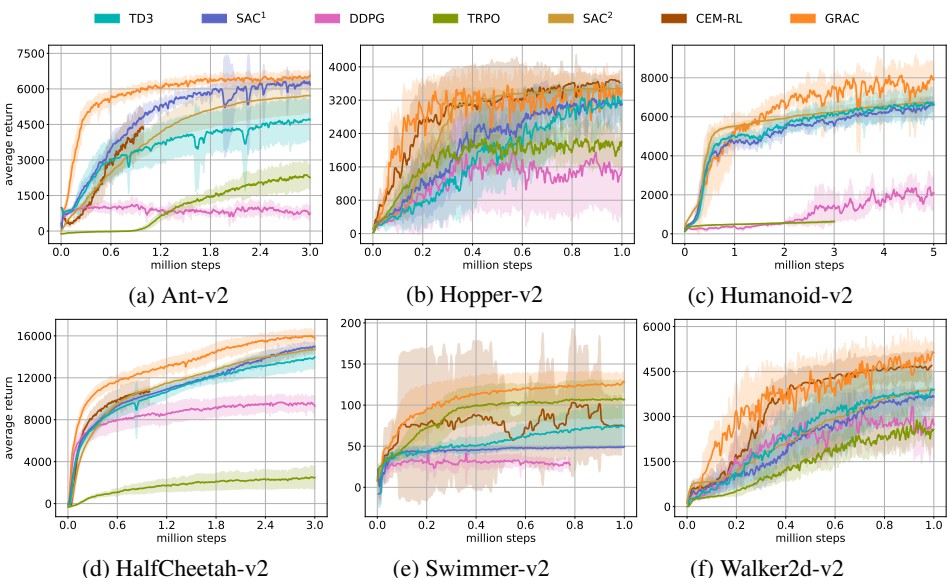

(a) Ant-v2  (b) Hopper-v2  (c) Humanoid-v2

(d) HalfCheetah-v2  (e) Swimmer-v2  (f) Walker2d-v2

Figure 1: Learning curves for the OpenAI gym continuous control tasks. For each task, we train 10 instances of each algorithm, using 10 different seeds. Evaluations are performed every 5000 interactions with the environment. Each evaluation reports the return (total reward), averaged over 10 episodes. For each training seed, we use a different seed for evaluation, which results in different start states. The solid curves and shaded regions represent the mean and standard deviation of the average return over 10 seeds. We do not apply any smoothing to the learning curves. *GRAC* (orange) learns faster than other methods across all tasks. *GRAC* achieves comparable result to the popular methods on the Hopper-v2 task and the Ant-v2 task and outperforms prior methods on the other four tasks including the complex high-dimensional Humanoid-v2.

## 5.1 Comparative Evaluation

We present *GRAC*, a self-guided and self-regularized actor-critic algorithm as summarized in Algorithm 1. To evaluate *GRAC*, we measure its performance on the suite of MuJoCo continuous control

tasks [26], interfaced through OpenAI Gym [11]. We compare our method with *DDPG* [5], *TD3* [1], *CEM-RL* [9], *TRPO* [4], $SAC^1$ [2], and $SAC^2$ [27]. We use the source code released by the original authors and adopt the same hyperparameters reported in the original papers and the number of training steps according to $SAC^1$ [2]. For *CEM-RL* [9] and $SAC^2$ [27], we use the results provided by their corresponding authors. $SAC^2$ does not contain results on Swimmer-v2. *CEM-RL* does not contain results on Humanoid-v2 and only runs one millions step on every tested task. Hyperparameters for all experiments are in the Appendix. Results are shown in Figure 1. *GRAC* outperforms or is comparable to all other algorithms in both final performance and learning speed across all tasks. On complex tasks with high state and action dimensions such as Humanoid-v2, *GRAC* outperforms all other algorithms by a large margin. Both *CEM-RL* and *GRAC* leverage CEM. *CEM-RL* applies *CEM* within the actor parameter space which is extremely high-dimensional while *GRAC* utilizes *CEM* within the action space of the Q function. *GRAC* outperform *CEM-RL* on tasks such as Ant-v2, HalfCheetah-v2, Swimmer-v2 by a large margin.

## 5.2    Ablation Study

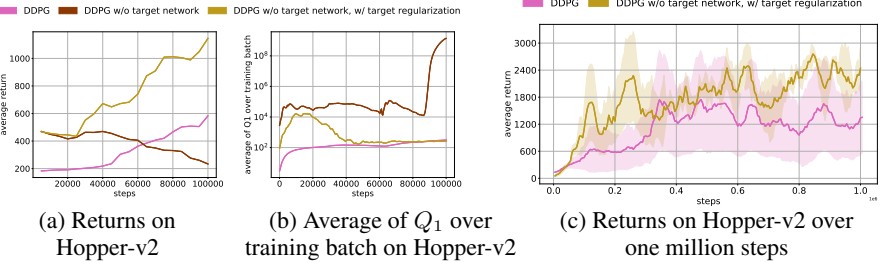

| (a) Returns on Hopper-v2 | (b) Average of $Q_1$ over training batch on Hopper-v2 | (c) Returns on Hopper-v2 over one million steps |

Figure 2: Learning curves and average $Q_1$ values ($y_1'$ in Alg. 1) on Hopper-v2. *DDPG* w/o target network quickly diverges as seen by the unrealistically high Q values. *DDPG* is stable but progresses slowly. If we remove the target network and add the proposed target regularization, we both maintain stability and achieve faster learning than *DDPG*.

In this section, we present ablation studies to understand the contribution of each proposed component: Self-Regularized TD-Learning (Section 4.1) and Self-Guided Policy Improvment (Section 4.2). We present our results in Fig. 4 in which we compare the performance of *GRAC* with alternatives, each removing one component from GRAC. Additional learning curves can be found in the Appendix. We also run experiments to examine how sensitive GRAC is to some hyperparameters such as $\alpha$ and $K$ listed in Alg. 1, and the results can be found in the Appendix.

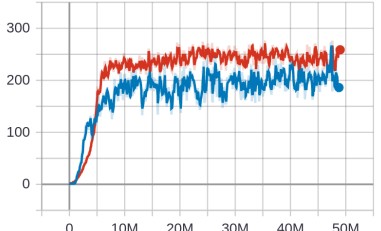

Figure 3: Average returns for the BreakoutNoFrameskip-v4 environment on OpenAI gym. Blue is **DQN**, red is **DQN w/o target network w/ target regularization**.

**Self-Regularized TD Learning**    To verify the effectiveness of the proposed self-regularized TD-learning method, we apply our method to *DDPG* (DDPG w/o target network w/ target regularization). We compare against two baselines: the original *DDPG* and *DDPG* without target networks for both actor and critic (*DDPG w/o target network*). We choose DDPG, because it does not have additional components such as Double Q-Learning, which may complicate the analysis of this comparison.

In Fig. 2, we visualize the average returns, and average $Q_1$ values over training batches ($y_1'$ in Alg.1). The $Q_1$ values of *DDPG w/o target network* changes dramatically which leads to poor average returns. *DDPG* maintains stable Q values but makes slow progress. Our proposed *DDPG* w/o target network w/ target regularization maintains stable Q values. In addition, we compare the average returns of *DDPG w/o target network, w/ target regularization* and *DDPG* within one million steps over four random seeds. *DDPG w/o target network, w/ target regularization* outperforms *DDPG* by large margins in five out of six Mujoco tasks (Fig. 2 shows results on Hopper-v2. The remaining results can be found in the Appendix). We also apply self-regularized TD-learning to DQN called *DQN w/o target network w/ target regularization* on the Atari Breakout environment and it outperforms *DQN* by 25%. The learning curve over 50 million steps is shown in Fig.3. These results on two different Q-learning methods demonstrate the effectiveness of our method and its potentials to be applied to a wide range of DRL methods.

**Policy Improvement with Evolution Strategies**    The GRAC actor network uses a combination of two actor loss functions, denoted as *QLoss* and *CEMLoss*. *QLoss* refers to the unbiased gradient estimators which extend the *DDPG*-style policy gradients [5] to stochastic policies. *CEMLoss* represents the policy improvement guided by the action found with the zero-order optmization method CEM. We run another two ablation experiments on all six control tasks and compare it with our original policy training method denoted as *GRAC*. As seen in Fig. 4, in general *GRAC* achieves a better performance compared to either using *CEMLoss* or *QLoss*. The significance of the improvements varies over the six control tasks. For example, *CEMLoss* plays a dominant role in Swimmer while *QLoss* has a major effect in HalfCheetah. This suggests that the *CEMLoss* and *QLoss* are complementary.

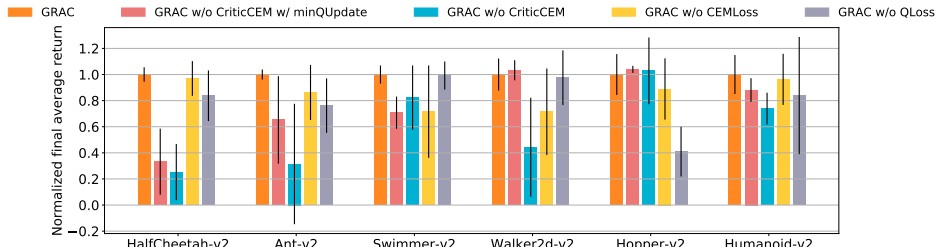

Figure 4: Final average returns, normalized w.r.t *GRAC* for all tasks. For each task, we train each ablation setting with 4 seeds, and average the last 10 evaluations of each seed (40 evaluations in total). The black lines represent one standard deviation. Actor updates without CEMLoss (*GRAC w/o CEMLoss*) and actor updates w.r.t minimum of both Q networks (*GRAC w/o CriticCEM w/ minQUpdate*) achieves slightly better performance on Walker2d-v2 and Hopper-v2. GRAC achieves the best performance on 4 out of 6 tasks, especially on more challenging tasks with higher-dimensional state and action spaces (Humanoid-v2, Ant-v2, HalfCheetah-v2). This suggests that individual components of GRAC complement each other.

## 5.3   In Hand Manipulation

We evaluate our approach on the problem of in-hand manipulation which remains unsolved due the high dimensionality of the problem and the complexity of multi-contact control [28]. Having this capability would allow robots to perform sophisticated tasks requiring repositioning and reorienting of grasped objects. We apply GRAC to a non-anthropomorphic robotic hand with 9 degrees of freedom [29]. We train the robotic hand in simulation to rotate a cube by 50 degrees around the gravity direction. The input states are the current, previous, initial, and target object position and orientation, and all nine gripper joint positions at the current time step. The actions are nine joint positions for the gripper at the next time step. GRAC learns a policy successfully accomplishing the task within 500k iterations. We test the learned policy on the real hand. Videos are included in the supplementary material.

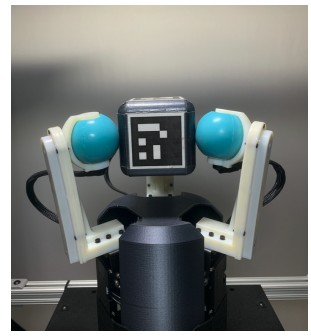

Figure 5:  Real-world Experiment Setup

## 6   Conclusion

Leveraging neural networks as function approximators, DRL has been successfully demonstrated on a range of decision-making and control tasks. However, the nonlinear function approximators also introduce issues such as divergence and overestimation. As our main contribution, we proposed a self-regularized TD-learning method to address divergence without requiring a target network that may slow down learning progress. The proposed method is agnostic to the specific Q-learning method and can be added to any of them. In addition, we propose self-guided policy improvement by combining policy-gradient with zero-order optimization such as the Cross Entropy Method. This helps to search for actions associated with higher Q-values in a broad neighborhood and is robust to local noise in the Q function approximation. Taken together, these components define *GRAC*, a novel self-guided and self-regularized actor critic algorithm. We evaluate *GRAC* on the OpenAI gym tasks, outperforming popular methods such as *TD3* [1] and *SAC* [2] on four tasks and achieving competitive results on two environments. We also run *GRAC* to enable a non-anthropomorphic robotic hand to successfully accomplish an in-hand manipulation task in the real world.

**Acknowledgments**

This work is supported by Toyota Research Institute (TRI). It is also supported by Siemens and the HAI-AWS Cloud Credits. We thank reviewers who gave useful comments.

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
