# OpenReview forum: "GRAC: Self-Guided and Self-Regularized Actor-Critic"
_robot-learning.org/CoRL/2021/Conference — CoRL2021 Poster_

### Official Review · Reviewer_8B1m · 2021-07-07

**Originality:** Good
**Technical Quality:** Very Good
**Clarity Of Presentation:** Very Good
**Impact:** 3

**Recommendation:**

Weak Accept: I recommend accepting the paper, but will not argue for my recommendation if the majority of other reviewers have a different opinion.

**Summary:**

This paper presents a set of deep reinforcement learning techniques that could be exploited to improve the performance of deep reinforcement learning algorithms. The authors propose two techniques.
The first technique is the safe regularized TD learning, which consists of an added regularization term, preventing the network to change considerably the next state best q value. This technique frees from the necessity of having a target network.
The second technique is  Self-Guided Policy Improvement with Evolution Strategies, which consists of optimizing the current policy using local search with CEM, instead of using policy gradient updates. This technique instead seems to speed up the maximization of policy actions wrt the critic functions. This effect could be greatly beneficial in actor-critic methods, where the ideal condition is that both value function and policy are consistent, i.e. we want the q function to have a value close to the policy value and the policy close to being optimal w.r.t. the value function. The authors also claim that makes the update more robust to noise, which is also desirable in this context.
Finally, the authors combine these two proposed techniques with Max-min Double Q-Learning to create GRAC, a novel deep RL algorithm. This algorithm shows very good performances in standard RL tasks. Authors also show that their approach is able to learn in simulation a manipulation task and transfer the resulting policy to the real robot.

**Issues:**

- Do a proper experimental campaign on Mujoco tasks (except Humanoid, which is quite heavy computationally) with 25 seeds for every algorithm. The computation time for running almost every State-of-the-art approach in these environments is roughly 1 day per environment, so it's totally doable, with a normal university cluster.
- Remove the rolling window cleanup from the graphs, it's not needed if you use a sufficient amount of seeds. Also, it could hide some instability of the method.
- Complete a proper evaluation for the CEM baseline. It's unacceptable to see a cut line in a graph. I would rather remove the comparison from the available ones than leaving this approach with incomplete experiments.
- In line 41, remove the "But" conjunction after the point. Use another more appropriate word.

**Reviewer Expertise:**

Excellent: Expert knowledge on the topic of the paper

**Strengths And Weaknesses:**

This paper proposes two very interesting techniques, that can be useful in Deep RL, particularly in the actor-critic settings.
The proposed algorithm seems to be sufficiently clean and easy to implement. The contribution looks very promising from the machine learning point of view.
The proposed ablation study is greatly executed and proves the claims of the paper and the advantages and complementarities of both methods.
the author proposes also a couple of interesting theorems on the optimality of the updates, even if the theoretical analysis should be extended (maybe in future works)

The major weakness of this paper is the experimental evaluation. The robotic experiment is a bit oversimplified, but I still think that the authors can prove the claim that this approach could be useful in a robotics environment.
My main concern is the Mujoco environment evaluation. After Henderson et. al "Deep Reinforcement Learning that Matters", it's unacceptable to see deep reinforcement learning experiments proposing new algorithms with less than 10 seeds per evaluation. While in the ablation study I can accept a more reduced statistical evaluation, it's unacceptable to have a novel algorithm with less than 10 seeds. Given that the tasks (excluding humanoid) are pretty easy to solve with moderate computational power, I suggest running 25 seeds to compute mean and confidence intervals.

**Summary Of Recommendation:**

The paper has very good potential if a proper experimental campaign with statistically relevant experiments is performed.
The common excuse "in other works few seeds are used" is not acceptable, as it's important to have a solid evaluation in deep learning approaches to have solid progress in this field.
The algorithm looks very promising and tackles important problems of deep reinforcement learning.
While I don't think that these RL approaches can be used to learn directly on a robot, I still think that the authors proved that these methods could be used to deploy a policy on the real system.
Currently, I propose a rejection, however, I'm willing to drastically change my recommendation if a proper experimental campaign is conducted. This can be easily achieved during the rebuttal period, as for most of the environments the computation time is very reduced.

After the response of the authors, I'll raise my evaluation to "weak accept", as the experimental evaluation has been/will be fixed.

---

### Official Review · Reviewer_wYEU · 2021-07-20

**Originality:** Fair
**Technical Quality:** Fair
**Clarity Of Presentation:** Fair
**Impact:** 3

**Recommendation:**

Weak Accept: I recommend accepting the paper, but will not argue for my recommendation if the majority of other reviewers have a different opinion.

**Summary:**

This paper presented several contributions including:
1. A self-regularized TD RL algorithm without conventional target network.
2. A self-guided policy improvement method robust to noise (using CME).
3. A Max-min Double Q-Learning introduced in 4.3

The proposed methods were tested in OpenAI gym tasks, although the relationship of each contribution is unclear. Section 5.3 presented a possible real task but the reviewer could not find any corresponding result.

**Issues:**

Please revise the paper following my comments in weaknesses.

**Reviewer Expertise:**

Very good: Comprehensive knowledge of the area

**Strengths And Weaknesses:**

Weaknesses
1. The link between each contribution is very weak.
   For the self-regularized TD approach, the reviewer wonders whether the regularization term will delay the learning？
   For the self-guided policy improvement method, it looks like a conventional policy based approach with CME, what is the contribution? What is the relationship between Theorems 1, 2 and the contribution of this paper? The CEM require the policy output a distribution of action, what is the policy network structure in GRAC?
   For the Max-min Double Q-Learning, what is its relationship with contributions 1 and 2?
2. No real robot experiment result in the paper. The results in attached video is too simple. The reviewer could not find how the proposed methods improved the learning performance in real robot experiment.

**Summary Of Recommendation:**

The current paper has a mess structure. The reviewer could not clearly understand the relationship between each proposed methods and their theoretical contribution. The real experiment does not have details so the reviewer could not understand its contribution in robot learning.

---

### Official Review · Reviewer_4Buh · 2021-07-24

**Originality:** Very Good
**Technical Quality:** Good
**Clarity Of Presentation:** Good
**Impact:** 4

**Recommendation:**

Weak Accept: I recommend accepting the paper, but will not argue for my recommendation if the majority of other reviewers have a different opinion.

**Summary:**

The authors introduce GRAC, an actor-critic RL framework that automatically regularizes the learning to prevent large spikes in Q value estimation (while obviating the need for target Q networks) and uses cross-entropy-guided search for improving action exploration economy. The proposed algorithm is theoretically justified and out-competes several current popular baselines on a number of OpenAI gym benchmarks.

**Issues:**

I imagine all the structural issues are fairly easy to address. Weaknesses W1-3 would also need to be addressed for me to consider improving my recommendation.

**Reviewer Expertise:**

Excellent: Expert knowledge on the topic of the paper

**Strengths And Weaknesses:**

### Strengths:

S1) As far as I can tell, this is indeed a novel framework and is theoretically compatible with any actor-critic implementation (though, realistically, it feels like it just stands on its own as an AC algorithm, given that adding blindly adding GRAC to existing AC algorithms could fundamentally break how they're meant to operate).

S2) The performance of the algorithm, in my opinion, does enough to justify it as another in the playing field.

S3) I appreciate the ablation study on the introduced elements as they help establish the value of the introduced algorithm elements.

S4) Overall, I think the paper maintains decent flow, barring a few structural issues (see below)

### Weaknesses:

W1) While a reasonable case can be made for the improved convergence speed of the proposed GRAC framework, it also introduces a number of additional computational elements - there are still multiple Q networks and there's the added compute burden of the CEM optimization of action choice. To make a claim of 'faster' training, I think it would be important to also show how compute cost changes under the proposed framework relative to baselines.

W2) A comparison of algorithms on the in-hand manipulation task seems to have been omitted but this omission was not justified in text. Is there some reason why the other algorithms could not be run on this task for a comparison? Given the relative performances on other tasks, I'd assume the other algorithms would have been able learn to at least somewhat handle this task if GRAC could.

W3) The authors claim in the abstract that GRAC is more robust to local noise in the Q-function but I felt this is never really quantitatively or theoretically substantiated.

W4) The authors claim Deep RL algorithms to be 'sample efficient' a number of times in the paper, but this is not a substantiated claim. While we can say that any one algorithm is less or more sample efficient than others, I do not believe we can make the claim yet that any deep RL methods are 'sample efficient' in an absolute sense - if anything, it's probably fair to say they're *not* sample efficient. I do not believe there has been a clear and comprehensive study on the sample efficiency of deep RL, but [1] and [2] do address it to a degree, as examples (this is honestly arguably more of a nitpick on semantics, but I feel like clarity is important for readers).

W5) The compared methods (TD3, SAC, DDPG, TRPO) are referred to as state-of-the-art, though they are relatively old now (by deep learning research standards), and newer methods have been developed and open-sourced, which perform better (eg. TQC [3]). I do not begrudge the authors their choice of baselines, but perhaps 'popular' is more appropriate a term than 'state of the art'.

#### Structural Issues:
SI1) It is unclear what you mean by gradient-based RL algorithms. Aren't all deep RL algorithms gradient-based, in a sense? Are you referring to policy gradients?

SI2) There is a lot of almost verbatim repetition in the text - eg. lines 64 - 68 and 79-80. It's fine to restate points, but at least paraphrasing them would help the flow - it was momentarily confusing to encounter verbatim sentences from earlier in the paper as it confused my sense of where I was in the text.

SI3) Missing '+' subscript in equation 7?

SI4) Line 199 and equation 9 -> change in notation from Q_1, Q_2 to Q(s,a,theta_1/2) feels unnecessary. It would read better to pick one format and stick to it.

SI5) It might be better to have the algorithm appear after the end of Section 4 as it would be easier to parse after the full description has been read.

SI6) Line 100: discount factor can also be 0 or 1 so \in [0,1] would be the more appropriate notation.

### Would be nice to have:

I realize it's rarely welcome when a reviewer asks for more data and/or analysis but it would be good to see how the regularization term's components (as introduced in section 4.1) change as training progresses with and without GRAC. It might help make a more visual case for the value of the regularization terms, though the ablation study in 5.2 is already perhaps sufficient, hence filing this as a 'would be nice to have'

#### References:
[1] Donner, F. Measuring Progress in Deep Reinforcement Learning Sample Efficiency, 2021

[2] Lee, S.Y. et al., Sample-Efficient Deep Reinforcement Learning via Episodic Backward Update, NeurIPS 2019

[3] Kuznetsov, A. et al., Controlling Overestimation Bias with Truncated Mixture of Continuous Distributional Quantile Critics, 2020 (also in stable baselines 3 contrib)

**Summary Of Recommendation:**

The paper has its merits, and as far as I can tell, is novel work that is likely competitive with current state of the art methods in RL. The lack of clear evaluation on the robotic task and various weaknesses listed do make me feel that it isn't yet what I'd consider a strong paper.

---

### Meta-Review · Area_Chair_GSUw · 2021-08-03

**Recommendation:** Accept (Poster)
**Confidence:** 4

**Metareview:**

=== comments before the discussion ===

The reviewers 4Buh and 8B1m agrees that the proposed method is novel. However, reviewers raised concerns related to experiments.
Especially, please address the following points:
- do experiments with some more random seeds and update the results (I would not request results with 25 random seeds, but 10 random seeds in total are necessary.)

- address the concerns related to experiments (please refer to comments from reviewers 4Buh and 8B1m)

Although reviewer wYEU criticized that the paper does not contain any real robot experiment, CoRL can accept  papers without real-robot experiments as long as papers try to address important problems for robotics.

=== comments after the discussion ===

This paper presents novel techniques for deep reinforcement learning. Authors improved the experimental results and addressed concerns raised by reviewers. The area chair recommends the acceptance of the paper.

---

### Decision · Program_Chairs · 2021-09-13

**Decision:**

Accept (Poster)

**Comment:**

=== comments before the discussion ===

The reviewers 4Buh and 8B1m agrees that the proposed method is novel. However, reviewers raised concerns related to experiments.
Especially, please address the following points:
- do experiments with some more random seeds and update the results (I would not request results with 25 random seeds, but 10 random seeds in total are necessary.)

- address the concerns related to experiments (please refer to comments from reviewers 4Buh and 8B1m)

Although reviewer wYEU criticized that the paper does not contain any real robot experiment, CoRL can accept  papers without real-robot experiments as long as papers try to address important problems for robotics.

=== comments after the discussion ===

This paper presents novel techniques for deep reinforcement learning. Authors improved the experimental results and addressed concerns raised by reviewers. The area chair recommends the acceptance of the paper.